# Measurement of Exhaled Volatile Organic Compounds as a Biomarker for Personalised Medicine: Assessment of Short-Term Repeatability in Severe Asthma

**DOI:** 10.3390/jpm12101635

**Published:** 2022-10-02

**Authors:** Adnan Azim, Faisal I. Rezwan, Clair Barber, Matthew Harvey, Ramesh J. Kurukulaaratchy, John W. Holloway, Peter H. Howarth

**Affiliations:** 1Clinical and Experimental Sciences, Faculty of Medicine, University of Southampton, Southampton SO16 6YD, UK; 2NIHR Southampton Biomedical Research Centre, University Hospital Southampton, Southampton SO16 6YD, UK; 3Department of Computer Science, Aberystwyth University, Aberystwyth SY23 3DB, UK; 4Human Development and Health, Faculty of Medicine, University of Southampton, Southampton SO16 6YD, UK; 5David Hide Asthma and Allergy Research Centre, Isle of Wight NHS Trust, Newport PO30 5TG, UK

**Keywords:** breathomics, VOC, volatile organic compounds, repeatability, asthma, severe asthma, respiratory

## Abstract

The measurement of exhaled volatile organic compounds (VOCs) in exhaled breath (breathomics) represents an exciting biomarker matrix for airways disease, with early research indicating a sensitivity to airway inflammation. One of the key aspects to analytical validity for any clinical biomarker is an understanding of the short-term repeatability of measures. We collected exhaled breath samples on 5 consecutive days in 14 subjects with severe asthma who had undergone extensive clinical characterisation. Principal component analysis on VOC abundance across all breath samples revealed no variance due to the day of sampling. Samples from the same patients clustered together and there was some separation according to T2 inflammatory markers. The intra-subject and between-subject variability of each VOC was calculated across the 70 samples and identified 30.35% of VOCs to be erratic: variable between subjects but also variable in the same subject. Exclusion of these erratic VOCs from machine learning approaches revealed no apparent loss of structure to the underlying data or loss of relationship with salient clinical characteristics. Moreover, cluster evaluation by the silhouette coefficient indicates more distinct clustering. We are able to describe the short-term repeatability of breath samples in a severe asthma population and corroborate its sensitivity to airway inflammation. We also describe a novel variance-based feature selection tool that, when applied to larger clinical studies, could improve machine learning model predictions.

## 1. Introduction

Asthma is characterised by chronic airway inflammation, variable airway obstruction/hyper-responsiveness, clinical symptoms (such as breathlessness and cough) and acute, potentially life-threatening exacerbation events [1,2]. Though most patients with asthma achieve good control with standard therapies, 5–10% of patients remain poorly controlled despite high-dose inhaled corticosteroid (ICS) and/or oral corticosteroids (OCS) [3]. It is increasingly recognised that these patients with severe asthma are highly heterogenous [4,5] and that there are distinct molecular mechanisms driving their poor asthma control [6,7,8,9] and varied response to currently available treatments [10,11,12].

The accurate identification of these driving mechanisms is critical in the ambition of precision medicine [13,14,15], and exhaled breath is potentially the ideal biomarker matrix for airways disease due to its direct contact with the organ of interest and ease of access. Exhaled breath can be sampled easily, non-invasively and almost inexhaustibly [16]. This is important because those biomarkers that can be easily sampled at the point of care have the highest translational potential [17], as illustrated by the clinical adoption of FeNO (fractional concentration of exhaled nitric oxide) for airway inflammation [18].

The primary components of exhaled breath are water vapour and inert gases, but the characteristic odours in breath are due to the thousands of volatile organic compounds present in tiny concentrations (parts per million to trillion) [19]. The measurement and high-throughput analysis of these volatile organic compounds, breathomics, has indicated that exhaled VOCs may be sensitive to airway inflammation [20,21].

Better understanding of exhaled VOCs is required before breathomics can be translated to the clinical setting [22]. Though within-day repeatability has been demonstrated in patients with asthma [23,24], VOCs are known to be sensitive to a number of non-disease-related factors liable to change day to day: exercise, diet and environmental exposures [20]. In this study, we sought to explore the short-term repeatability of VOCs in severe asthma patients during a clinically stable state. Understanding this quality is critical if exhaled VOCs are to be used as clinical biomarkers that can guide personalised medicine approaches.

## 2. Materials and Methods

### 2.1. Population

The WATCH (Wessex AsThma CoHort of difficult asthma) study [25] is an ongoing prospective clinical cohort of patients with difficult-to-treat asthma based at University Hospitals Southampton NHS Foundation Trust (UHSFT), Southampton, United Kingdom (UK). In this study, patients with severe asthma were invited for deeper clinical characterisation, which included the collection of blood, breath, and induced sputum samples. Participants were included in this analysis if they agreed to participate in the study schedule described below and were able to produce a viable sputum sample. Severe asthma was confirmed by an asthma specialist in accordance with the BTS (British Thoracic Society) guidelines with alternative causes for symptoms excluded and treatment for co-morbidities optimised. Participants were aged between 18 and 80 years, not current smokers but with no restrictions according to sex or race.

### 2.2. Study Design

The full characterisation schedule was performed on the same morning ending with sputum induction. In some cases, to obtain a viable sputum sample, it was necessary to repeat sputum induction on a second date. Subjects were excluded from the analysis if a viable sputum sample was not obtainable. Once a sputum sample was obtained, subjects were invited to provide breath samples on five consecutive days, starting within 7 days of sputum induction; these breath samples were collected at the same time of day.

### 2.3. Sputum Collection

Sputum was induced using a DeVilbiss^®^ Ultraneb (DeVilbiss, New York, NY, USA) following a standardised protocol based on the methods described by ten Brinke et al. [26]. Subjects were bronchodilated with short-acting beta-agonist (SABA) medication prior to sputum induction, and lung function (FEV_1_) was measured. Subjects underwent a maximum of 3 rounds of 5-min saline nebulisations, with increasing saline tonicity, beginning with isotonic (0.9%) saline, followed by 3% and finally 4.5% saline. To check tolerability of the procedure, lung function (FEV_1_) was measured after each 5-min nebulisation and after 2-min of nebulisation if the subject’s FEV_1_ < 1.5 L. If a 20% drop from post bronchodilator FEV_1_ had been reached, the induction would be stopped. Samples were stored on ice during collection and transport to the laboratory for processing.

### 2.4. Sputum Processing

The concurrent method of sputum processing was performed involving a phosphate buffered saline (PBS) incubation step followed by DTE incubation providing cytokines and a differential cell count, respectively [27]. Sputum samples were processed as soon as possible and within 2 h of expectoration with 8× volume of PBS, a proportion of supernatant was then removed, and the sample was further incubated with 0.2% dithioerythritol (DTE), giving a final concentration of 0.1% DTE before cytospins for cell counts were obtained. Cytospins were stained using by rapid Romanowski staining (Fisher Scientific, Loughborough, UK). The proportion of inflammatory cells were assessed by counting 800 respiratory cells plus squamous to give a mean percentage of respiratory cells.

### 2.5. Breath Sampling

All breath samples were collected within the same room. Breath samples were collected using the ReCIVA Breath Sampler (Owlstone Medical Ltd, Cambridge, UK). Exhaled breath was collected onto a Breath Biopsy Cartridge, which consists of four Tenax TA/Carbograph 5TD sorbent tubes (Markes International, Llantrisant, UK). The ReCIVA Breath Sampler monitored subjects’ tidal breathing pattern in real time, using CO_2_ concentration and pressure sensors. Dynamically determined gates using the real-time pressure levels triggered the sampling pumps to collect breath. Each pump pulls pressure-gated exhaled breath through two sorbent tubes, with 1473 mL being collected on each tube. Each pair of tubes was later combined to give a single sample for thermal desorption-gas chromatography-mass spectrometry (TD-GC-MS) analysis.

### 2.6. Breath Analysis

Samples were dry purged to remove excess water and desorbed using a TD100-xr thermal desorption autosampler (Markes International) and transferred onto a Quadrex 007-624 column (30 m × 0.32 mm × 3.00 µm) using splitless injection. Chromatographic separation was achieved via a programmed method (40–250 °C in 84.5 min at 3.0 mL/min) on a 7890B gas chromatography (GC) oven (Agilent Technologies, Santa Clara, CA, USA) and mass spectral data acquired using an electron impact ionization time-of-flight (TOF) BenchTOF high-definition mass spectrometer (MS) (Markes International). Each sample consisted of two sorbent tubes, both of which were desorbed into the Thermal Desorber cold trap for a single analysis. A cleaning method was run between each sample to prevent carry-over.

A quality control (QC) sample (sorbent tube spiked with a known mixture of chemicals) was run between every four subject breath samples to monitor the stability of instrumentation. A blank tube was run every four samples and after every quality control sample to monitor background. A set of four samples, quality control samples, and blank tubes are denoted as an “analytical sequence”.

### 2.7. Breath Data Pre-Processing

Retention time shifts due to column events were corrected using retention time of compounds in QC samples. For each QC sample, a piece-wise linear function was constructed by comparing QC compound retention times in the sample to the compound-specific medians across all QC samples. This piece-wise linear function was then applied to the retention time axis of breath samples that were analysed immediately after the QC sample. Small deviations in peak area, introduced by retention time alignment, was corrected using the scaling factors derived from the piece-wise linear functions.

Untargeted feature extraction was performed for samples that passed all curation checks. TD-GC-MS chromatograms were converted into molecular feature (MF) lists for statistical analysis. Whenever a feature was missing due to limit of detection (LOD), the baseline for that feature was integrated instead to give a minimum value. If a feature could not be reliably quantified due to issues not associated with LOD (e.g., interference from neighbouring peaks), no baseline integration was performed, and the feature was marked as non-LOD missing.

Any feature with a high frequency (>20% of all samples considered in feature extraction) of non-LOD missing values was excluded from further analysis. Each feature was assigned a tentative ID by comparison to the National Institute of Standards and Technology (NIST) mass spectra standard reference database (2017). A tentative ID was assigned if the match score was >85%.

### 2.8. Statistical Analysis

Statistical analysis was performed using Python scripting language (version 3.8.3) [28]. Clinical characteristics were described using median and 95% confidence intervals with between-group comparisons by Mann–Whitney U tests for continuous variables and absolute numbers with percentages within each group by Chi Squared tests for categorical variables.

The abundance of molecular features (used hereafter interchangeably with VOCs) was batch corrected (ComBAT [29]), log transformed and scaled (minmaxscaler). Data were explored using principal component analysis (PCA) and visualised using the first two principal components with ellipses constructed and positioned using the mean and coefficient of variation of the principal components within the grouping of interest as a representation of a 95% confidence interval. Correlations between principal components and clinical characteristics was done by Pearson’s correlation coefficient for continuous variables and by point biserial correlation for categorical variables. Sputum granulocyte percentages were log transformed prior to correlation.

The variation for each VOC was calculated using the median absolute deviation: median ratio (CV_MAD_), a coefficient of variation that performs well when applied to data of skewed distributions [30]. The “Within Subject” variation for a VOC was defined by calculating the CV_MAD_ across five samples in the same subject and averaged across all the subjects. The “Between Subject” variation for a VOC was defined by averaging the measure across five samples in the same subject and calculating the CV_MAD_ across all subjects. Any VOC with a mean between-subject variability of ≥30% was considered potentially discriminatory. Any VOC with a mean within-subject variability of ≥30% was considered inconsistent. These criteria were used to categorise VOCs into four: “Conserved”: low variability within subjects and between subjects, “Erratic”: high variability within subjects and between subjects, “Potential biomarkers”: low variability within subjects but high variability between subjects, and “Noisy”: high variability within subjects but low variability between subjects.

Unsupervised clustering was performed using a K Means algorithm and Ward Hierarchical algorithm on Euclidean distances. Both algorithms were instructed to identify *n* clusters, where *n* represents the number of subjects from which the breath samples were collected. Concordance between cluster prediction and subject identifier was assessed using the Adjusted Rand Index and Fowles Mallows Score.

## 3. Results

The subjects participating in this study had poorly controlled asthma (ACQ6 of 2.5) with evidence of persistent T2 airways inflammation (median FeNO of 38.5 ppb and median Sputum Eosinophils of 2.6%) despite high doses of ICS therapy. In contrast to established asthma cohorts, these subjects were predominantly male (35% female) and lean (median BMI 25.0). Other than sputum neutrophils, there was no statistically significant difference between males and females (Table 1).

### 3.1. Exploratory Analysis by PCA

In total, 32.75% of the variance in VOCs across the 70 breath samples (5 from each of the 14 subjects) was captured in two principal components (Figure 1A). A PCA plot using the first two principal components illustrated no separation of samples according to the day on which breath samples were collected (Figure 1D). Breath samples from the same subject broadly cluster closely to one another (Figure 1C) but do show some within-subject variability, as illustrated by the ellipses. The size of each ellipsis (representing an individual subject) relative to the spread of all breath samples illustrates that within-subject variability is a fraction of the variability seen across all breath samples. The ellipses are closely connected and in most cases overlap, indicating that breath samples from different individuals share some characteristics.

Correlation between clinical characteristics and VOC-derived principal components indicates that the majority of variance (first three PCs) in repeated VOC measures are most sensitive to the Subject’s identity (Figure 1B). Thereafter, the variance appears sensitive to characteristics associated with T2 inflammation: atopy, FeNO and sputum eosinophilia. None of the variance captured in the first 10 principal components (accounting for 76.9% of all variance in the VOCs) relates to the day of the visit.

### 3.2. Within-Subject Variability of Individual VOCs

The majority of VOCs (62, 69.66%) had a mean within-subject variation of <30% (Figure 2A). In total, 15.73% of VOCs (*n* = 14) were found to be “Conserved”, that is, they showed low variability within subjects and between subjects. None of the VOCs were categorised as “Noisy” but 30.35% of VOCs (*n* = 27) were found to be “Erratic”, that is, they showed high variability within subjects and between subjects. The remaining 53.93% of VOCs (*n* = 48) showed low variability within subjects but high variability between subjects, which we labelled “Potential Biomarkers” (based on these variability criteria alone).

The most frequent VOCs identified in breath samples in this cohort were classified as alkanes or terpenoids (Figure 2B). A total of 52.4% of alkanes met our variance-based criteria for a potential biomarker; however, 38.1% also met our criteria for erratic markers. The majority of aldehydes (57.1%) were erratic.

### 3.3. Impact of Removing Erratic Volatile Organic Compounds

Unsupervised clustering by hierarchical and k means clustering, instructed to predict 14 (number of subjects) clusters from the repeat breath samples (*n* = 70), showed good concordance between their predictions and actual subject identifiers, as measured by the Adjusted Rand Index (Figure 3A). There were no differences in scores between models trained on all features and models trained only on non-erratic features (Figure 3A). The silhouette score uses the mean intra-cluster distance and mean nearest-cluster distance to represent how distinct each cluster is. Using this metric, restricting to non-erratic features produced more distinct clusters for both clustering algorithms (Figure 3B). A PCA was repeated on non-erratic VOCs from all samples and correlated against clinical characteristics (Figure 3C), replicating Figure 1B but excluding erratic VOCs. A similar pattern to that observed when using all VOCs (Figure 1B) indicates that the underlying structure of VOCs has not been compromised by removing these erratic features: in particular, statistically significant correlations are observed between the first two principal components and FeNO, sputum eosinophils (%) and sputum neutrophils (%).

## 4. Discussion

Inflammatory phenotyping has become essential in the clinical assessment of asthma [3], identifying patients that will respond to corticosteroid and IL-5 targeting therapies [31]. Whilst high-throughput molecular analyses of sputum samples demonstrates that there is heterogeneity beyond phenotyping by inflammatory cell counts alone [7,32], sampling this matrix is unsuitable for routine clinical practice or large epidemiological studies due to the practical limitations of undertaking sputum induction [33]. Validation of the reliable identification of these molecular mechanisms through non-invasive technologies, such as breathomics, is critical in advancing precision medicine in severe asthma.

To our knowledge, this is the first report of the short-term repeatability of exhaled VOCs in a severe asthma cohort. During a clinically stable state, the abundance of most VOCs included in this analysis were consistent day to day. This form of analytical validity is essential if breathomics are to become established as clinical biomarkers and used in clinical decision making. Alkanes and terpenoids are commonly identified as discriminatory in exhaled breath [34] and, in this study, were broadly identified as having low variance within subjects and high variance between subjects, giving confidence to their application as a clinical biomarker.

The primary source of variation on PCA in the present analysis is subject identity. Beyond this, some caution should be used in interpreting correlation with clinical characteristics, as the PCA was performed in repeated measures. Nevertheless, the results indicate that PC1 is also sensitive to T2 biology. This is consistent with other breathomics studies [21], including those employing PCA in which the first PC usually discriminates airways disease from health but thereafter variance relates to airway inflammation [35]. PC1 on this data also correlated with sex, which is well recognised to influence exhaled VOCs [36]. Though our study was limited by numbers, none of the other clinical characteristics associated with PC1 were statistically significantly different between males and females to indicate that sex underlies the sensitivity to T2 biology. Similarly, correlations to subsequent PCs were not shared with sex.

The sampling parameters in this analysis may not match those of other studies and so, strictly, the variance-based categorisation of VOCs described here may not apply to the same VOCs measured in other studies. This ultimately reinforces the need for methodological standardisation [37,38]. Nevertheless, the application of within-subject variance as a feature selection tool is, to our knowledge, unique. Dimension reduction techniques are commonly applied to breathomics [39] and have a number of advantages. In particular, when the number of observations is limited, dimension reduction has the advantage of reducing model overfitting and in so doing, increasing overall performance of machine learning classifiers [40].

It is plausible that in some instances, within subject variance may reflect salient changes in airway biology; penalising this variation may therefore ignore or limit the potential of breathomics. However, this consideration should be restricted to established VOC biomarkers. The pressing need for breathomics is validation, but conducting large-scale studies with enough observations to overcome the risk of false discovery is logistically [41] and financially costly and thus prohibitive. One way to augment this process may be to conduct a repeatability study, as described here, in parallel to a larger cross-sectional study: VOCs that are erratic are most likely to introduce noise to data and contribute to false positives. As the field grows, it is likely that this approach is superseded.

The impact of this feature selection tool was assessed on unsupervised clustering tools predicting subject identifier, as our data consisted of repeat samples and was underpowered for the prediction of clinical characteristics. Subject identity was a major contribution to variance in this data and so it is not surprising that there is a ceiling effect for classification using all or restricted features, as measured by the adjusted Rand index. Improvement in the silhouette score does indicate though that the clusters are more distinct when using a restricted set of features. Future research should investigate the value of variance-based feature selection on supervised machine learning models predicting clinically informative parameters. Our findings indicate that these models might be improved by this technique.

## 5. Conclusions

In summary, we demonstrate that the majority of VOCs are consistently measured in severe asthma patients over a short-term period of stability. We demonstrate that the exclusion of erratic VOCs improves the performance of machine learning models without sacrificing salient discriminatory data, including sensitivity to underling airway inflammation, and may represent a novel adjunct to future trial design.

## Figures and Tables

**Figure 1 jpm-12-01635-f001:**
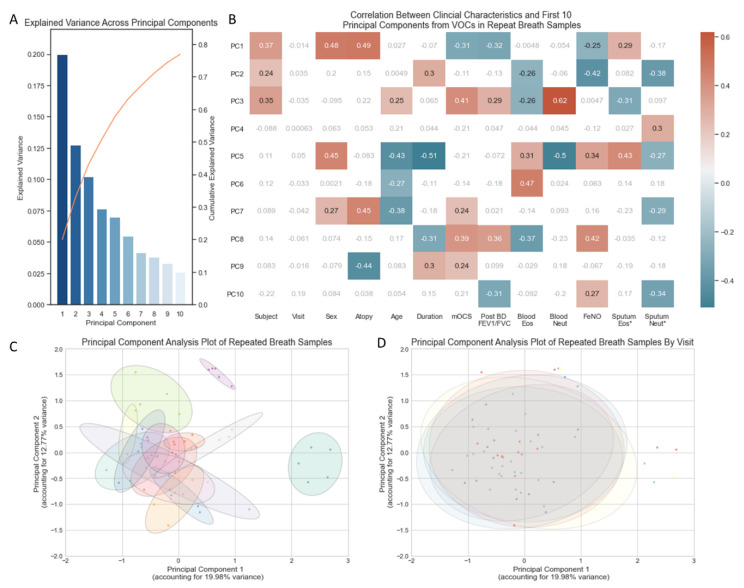
(**A**) Bar chart of the explained variance captured by each of the first ten principal components. (**B**) Heatmap of correlation between clinical characteristics and the first 10 principal components (Appendix A), non-significant correlations in grey/white, positive and negative correlations with *p*-value < 0.05 in red and blue, respectively. * Sputum Eosinophils and Sputum Neutrophil Percentages log transformed. (**C**) PCA plot of all 70 breath samples with ellipses representing subject identifiers (*n* = 14). (**D**) PCA plot of all 70 breath samples (5 samples from 14 subjects) with ellipses representing the day of the week (Monday, Tuesday, Wednesday, Thursday, Friday) on which the sample was collected. Volatile organic compounds (VOCs), principal component (PC), maintenance oral corticosteroid treatment dose (mOCS), post bronchodilator ratio between forced expiratory volume in 1 s and forced vital capacity (Post BD FEV1/FVC), eosinophils (Eos), neutrophils (Neut), Fraction of exhaled Nitric Oxide (FeNO).

**Figure 2 jpm-12-01635-f002:**
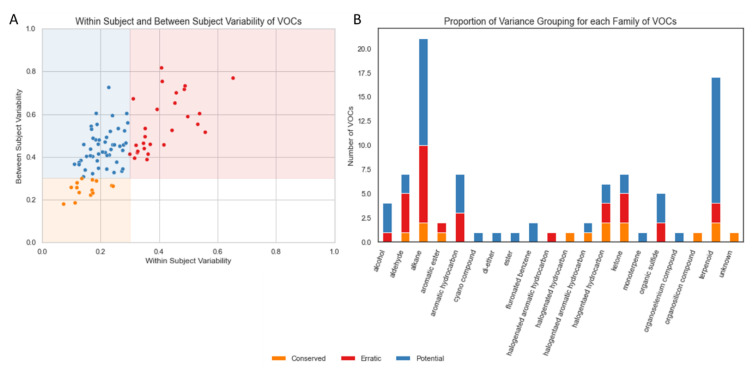
(**A**) Scatterplot of within-subject variability and between-subject variability for each VOC. (**B**) Barplot indicating the frequency of VOC type observed in the NIST (National Institute of Standards and Technology) -identified VOCs used in this analysis, coloured by the relative frequency of repeatability categories derived from within- and between-subject measures. Volatile organic compounds (VOCs), National Institute of Standards and Technology (NIST).

**Figure 3 jpm-12-01635-f003:**
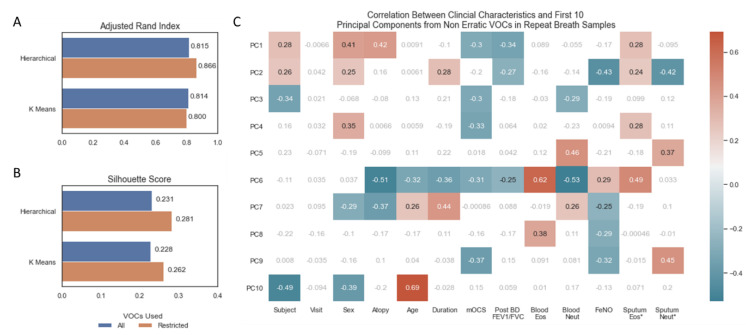
(**A**) Barplot of adjusted rand index for assessing agreement in unsupervised machine learning algorithm prediction of 14 clusters and 14 subject identifiers. (**B**) Fowles Mallows Score for assessing agreement in unsupervised machine learning algorithm prediction of 14 clusters and 14 subject identifiers. (**C**) Heatmap of correlation between continuous clinical characteristics and the first 10 principal components, non-significant correlations in grey/white, positive and negative correlations with *p*-value < 0.05 in red and blue, respectively. * Sputum Eosinophils and Sputum Neutrophil Percentages log transformed. Volatile organic compounds (VOCs), principal component (PC), maintenance oral corticosteroid treatment dose (mOCS), post bronchodilator ratio between forced expiratory volume in 1 s and forced vital capacity (Post BD FEV1/FVC), eosinophils (Eos), neutrophils (Neut), fraction of exhaled nitric oxide (FeNO).

**Table 1 jpm-12-01635-t001:** Cohort characteristics by sex. Body mass index (BMI), gastro oesophageal reflux disorder (GORD), inhaled corticosteroids (ICS), oral corticosteroids (beclomethasone dipropionate equivalent) (OCS) immunoglobulin E (IgE), interleukin—(IL-5), fraction of exhaled nitric oxide (FeNO), parts per billion (ppb), forced expiratory volume in 1 s (FEV1), ratio between forced expiratory volume in 1 s and forced vital capacity (FEV_1_/FVC), measure relative to predicted value (% pred), asthma control questionnaire (ACQ6), hospital anxiety and depression scale questionnaire (HADS), Sino-nasal outcome test (SNOT22). Values displayed as number (%) or median [Q1, Q3].

	Missing	Overall	Male	Female	*p* Value
		14	9	5	
Sex (% Female)	0	5 (35.7)	0	5 (100.0)	<0.001
Age (years)	0	54.0 [51.2, 60.2]	58.0 [52.0, 67.0]	51.0 [39.0, 54.0]	0.094
Age of Onset (years)	4	14.0 [5.2, 30.2]	14.0 [4.8, 24.8]	21.5 [9.8, 31.2]	0.669
Atopic	0	8 (57.1)	7 (77.8)	1 (20.0)	0.091
Smoker (% never)	0	11 (78.6)	7 (77.8)	4 (80.0)	1
BMI (kg/m^2^)	0	25.0 [23.3, 31.5]	25.1 [24.1, 32.4]	23.9 [23.1, 27.0]	0.386
Nasal Polyps	1	6 (46.2)	4 (50.0)	2 (40.0)	1
GORD	0	11 (78.6)	7 (77.8)	4 (80.0)	1
ICS therapy dose	0	2920.0 [2529.0, 3900.0]	2920.0 [2460.0, 3000.0]	3840.0 [2920.0, 3920.0]	0.459
Maintenance therapy OCS	0	5 (35.7)	4 (44.4)	1 (20.0)	0.58
Anti IgE therapy	0	0	0	0	1
Anti IL-5 therapy	0	3 (21.4)	2 (22.2)	1 (20.0)	1
Asthma Exacerbations requiring OCS (last 12 months)	1	1.0 [0.0, 3.0]	0.5 [0.0, 1.0]	3.0 [1.0, 4.0]	0.194
FeNO_50_ (ppb)	0	38.5 [29.8, 50.8]	40.0 [29.0, 50.0]	32.0 [32.0, 65.0]	0.841
Post BD FEV_1_ % pred	0	81.5 [45.6, 92.9]	85.4 [44.0, 92.7]	77.6 [50.5, 99.7]	0.641
Post BD FEV_1_/FVC % pred	0	66.0 [54.5, 74.5]	66.0 [62.0, 70.0]	64.0 [52.0, 81.0]	0.841
ACQ6	0	2.5 [1.6, 3.5]	2.7 [1.3, 3.7]	2.3 [2.3, 2.8]	0.841
HADS Score	1	11.0 [6.0, 14.0]	11.0 [6.0, 20.5]	11.0 [6.0, 12.0]	0.462
Blood Eosinophils	0	0.2 [0.1, 0.4]	0.2 [0.1, 0.4]	0.3 [0.2, 0.3]	1
Blood Neutrophils	0	4.7 [4.2, 6.2]	5.5 [4.6, 7.3]	4.2 [3.7, 4.3]	0.062
Serum Total IgE	0	221.8 [52.9, 386.4]	234.0 [104.9, 367.4]	209.6 [35.5, 392.7]	0.841
Sputum Eosinophils (%)	0	2.6 [0.3, 22.5]	1.8 [0.5, 5.8]	40.9 [0.2, 47.0]	0.385
Sputum Neutrophils (%)	0	39.8 [17.8, 68.0]	65.0 [54.6, 77.5]	17.4 [11.1, 24.0]	0.028

## Data Availability

The data presented in this study are available on request from the corresponding author.

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
