# Peer review of "Measurement of Exhaled Volatile Organic Compounds as a Biomarker for Personalised Medicine: Assessment of Short-Term Repeatability in Severe Asthma"

_jpm, 2022, doi:10.3390/jpm12101635_

Round 1

Reviewer 1 Report

Thank you for an interesting paper. I have a few questions.

1. My major question is the number of study subjects, beeing only 14. How did you select them from the WATCH- cohort and could you please include information on how many were excluded and for what reason (only not obtaining sputum or were there other drop-outs).

2. Table 1. Please give information on ICS therapy dose (Budesonide equivalent?). What information does SNOT22 add. You are lacking that in 1/3 of the participants and no conclusion should thereby be drawn on the value for the whole group.

Figure 1 and 3. Are all the first 10 Principal Components "Potential biomarkers". If so is the case try to explain differerences in figure 1B and 3 C, were some show positive association in figure 1 and negative in figure 3. If not, which PC are the same in the two Figures.

In figure 3 you refere to A, B , C and D in capture text, but only include 3 figures.

In conclusion I agree that you can say that the majority of VOCs are consistent over the time sampled, but can you draw the conclusion that a majority appear sensitve to underlying airway inflammation from your limited sample size andthe somewhat diverging results from Figure 1 and 3?

Overall, thanks again for this paper and it adds to the information on low shortterm within subject variability of the examined VOCs in subjects with severe asthma.

Author Response

Dear Reviewer, many thanks for your interest in our manuscript and helpful comments. We have addressed each point below and believe the revisions have enhanced the manuscript as a result. 

Point 1: My major question is the number of study subjects, beeing only 14. How did you select them from the WATCH- cohort and could you please include information on how many were excluded and for what reason (only not obtaining sputum or were there other drop-outs).

Response 1:  Many thanks. The primary limiting factor to recruitment was commitment to the study schedule. I have added clarification on selection criteria in the Methods section

Point 2: Please give information on ICS therapy dose (Budesonide equivalent?).

Response 2: Apologies, the ICS dose is expressed as beclomethasone dipropionate equivalent, which has now been clarified in the table caption.

Point 3: What information does SNOT22 add. You are lacking that in 1/3 of the participants and no conclusion should thereby be drawn on the value for the whole group.

Response 3: Agree, this is a valid limitation. SNOT22 has therefore been removed from the table.

Point 4: Figure 1 and 3. Are all the first 10 Principal Components "Potential biomarkers". If so is the case try to explain differerences in figure 1B and 3 C, were some show positive association in figure 1 and negative in figure 3. If not, which PC are the same in the two Figures.

Response 4: The intention is to describe similarity between 1B and 3C despite removing a proportion of VOCs – I have added clarification in the text.

Point 5: In figure 3 you refere to A, B , C and D in capture text, but only include 3 figures.

Response 5: Apologies – I have corrected this.

Point 6: In conclusion I agree that you can say that the majority of VOCs are consistent over the time sampled, but can you draw the conclusion that a majority appear sensitve to underlying airway inflammation from your limited sample size and the somewhat diverging results from Figure 1 and 3?

Response 6: Thank you. No, I agree that we cannot conclude that the majority of VOCs are sensitive to underlying airway inflammation – I have removed this erroneous assertion from the conclusion.

Point 7: Overall, thanks again for this paper and it adds to the information on low shortterm within subject variability of the examined VOCs in subjects with severe asthma.

Response 7: Many thanks

Reviewer 2 Report

In this manuscript, Adnan Azim et al. measured exhaled VOCs from severe Asthma patients in a short-term repeated way, to predict biomarkers for personalized medicine. After PC analysis of VOCs, the authors found that PC1, 2, and 3 significantly correlated with the subjects, partially associated with the lung physical function such as FEV1/FVC, and partially correlated with the local type 2 inflammatory cell types. Moreover, excluding erratic VOCs from machine learning approaches didn’t alter the correlation too much. I have some substantial concerns about this manuscript:

1.     The authors used PCA for the VOCs grouping and dimension reduction. I wonder why the authors didn’t use tSNE or UMAP to reduce the dimension; these methods are more potent for this aim and may give authors better separation.

2.     Could the authors list the contents of each PCs as the supplementary table?

Author Response

Dear Reviewer, many thanks for your interest in our manuscript and helpful comments. We have addressed each point below and believe the revisions have enhanced the manuscript as a result.

Point 1: The authors used PCA for the VOCs grouping and dimension reduction. I wonder why the authors didn’t use tSNE or UMAP to reduce the dimension; these methods are more potent for this aim and may give authors better separation

Response 1:  Many thanks for this question. The reason for using PCA over UMAP was two-fold. Firstly, t-SNE and UMAP do not allow interrogation or interpretation of source features, which we felt important for our analysis. Secondly, whilst the maximising of local structure on dimension reduction through non-linear techniques, such as t-SNE and UMAP, has clear advantages when prioritising separation for visualisation, the purpose of this analysis was to additionally illustrate similarities between samples. PCA preserves global structure and so, in the original Figure 1 C, we see some overlap between breath samples from different patients, an observation that is lost when using UMAP. We describe this characteristic as part of the results section and believe that this is important to maintain.

Point 2: Could the authors list the contents of each PCs as the supplementary table?

Response 2: I have interpreted “contents of each PC” as the loadings: how each VOC contributes to each of the first 10 Principal Components: I included this table in the supplementary table.

Round 2

Reviewer 2 Report

I'm satisfied with the revision.